# Production and characterization of bio-oils from fast pyrolysis of tobacco processing wastes in an ablative reactor under vacuum

**Nattawut Khuenkaeo[1], Sanphawat Phromphithak[1], Thossaporn Onsree[1], Salman Raza Naqvi[2], Nakorn Tippayawong[1]***

**1** Faculty of Engineering, Department of Mechanical Engineering, Chiang Mai University, Chiang Mai, Thailand, **2** Department of Chemical Engineering, National University of Sciences and Technology, Islamabad, Pakistan

* n.tippayawong@yahoo.com

**Data Availability Statement:** All relevant data are within the manuscript and its Supporting Information files.

## Abstract

Application of advanced pyrolysis processes to agricultural waste for liquid production is gaining great attention, especially when it is applied to an economic crop like tobacco. In this work, tobacco residues were pyrolyzed in an ablative reactor under vacuum. The maximum bio-oil yield of 55% w/w was obtained at 600°C with a particle size of 10 mm at a blade rotation speed of 10 rpm. The physical properties of the products showed that the oil produced was of high quality with high carbon, hydrogen, and calorific value. Two-dimensional gas chromatography/time-of-flight mass spectrometric analysis results indicated that the oils were complex mixtures of alkanes, benzene derivative groups, and nitrogen-containing compounds. In addition, $^{13}$C NMR results confirmed that long aliphatic chain alkanes were evident. The alkanes were likely converted from furans that were decomposed from hemicelluloses. Ablative pyrolysis under vacuum proved to be a promising option for generating useful amount of bio-oils from tobacco residues.

## 1. Introduction

Due to the depletion of fossil fuel supplies, along with major environmental concerns, renewable energy sources such as solar, wind, hydro, geothermal, and biomass are becoming enormously important. Among these sources, biomass has drawn attention because of its possible transformation into valuable oil products. Agro-waste is one of the most significantly exploited renewable energy sources; for example, it is used to produce fuels and valuable chemicals [1]. In the north of Thailand, tobacco is a major economic crop, and harvesting and processing it generates large amounts of waste, which is toxic if managed poorly. However, this waste can be used to generate valuable and useful products via pyrolysis [2].

Pyrolysis is a thermal cracking process for biomass feedstock without or with a limited supply of oxygen. It usually operates in a temperature range of 300–700°C. Liquid oil, solid char, and gas are three major products of the pyrolysis process [3–6]. A thermal reactor that provides heating is a major part of the process. Ablation is about pressing a material against hot surface with high relative motion. It is a unique rapid heating technique that enables direct heat transfer between relatively large sized biomass and heat source, consequently the rigorous

**Funding:** NT received funding from the National Research Council of Thailand (www.nrct.go.th), and Chiang Mai University (www.cmu.ac.th). NK and SP received RA scholarships from CMU Graduate School (www.grad.cmu.ac.th) The funders had no role in study design, data collection and analysis, decision to publish, or preparation of the manuscript.

**Competing interests:** The authors have declared that no competing interests exist.

reduction in feed size is not required. Ideally, the pyrolytic volatiles generated need to be rapidly removed from the reactor and quenched, possibly in a condensing system, to avoid secondary cracking reactions and repolymerization, and formed bio-oil [5–7]. A vacuum pyrolysis system may be adopted without the need of carrier gas to generate valuable reactive intermediates [8,9]. The relative yield and characteristics of liquid product depends on the operating parameters and the properties of the biomass.

Biomass pyrolysis produces liquid bio-oil, which is a source of biochemicals and biofuels. Several works have pyrolyzed tobacco waste to obtain high-value products such as biochar [10,11], which investigated the effects of pyrolysis conditions on solid residue and the physiochemical properties of the char. Some researchers have investigated thermal decomposition [12,13], studying the potential of pyrolysis for the valorization of tobacco using thermogravimetric (TG) analysis and TG-Fourier transform infrared (FTIR) spectroscopic techniques to investigate the effect of temperature and heating rate on the products [14,15]. It is well known that slow pyrolysis results in high char yield, whereas high yields of liquid oil are always obtained from fast pyrolysis. Pütün et al. [16] investigated yields and chemical composition between slow and fast pyrolysis. They found that not only was the type of biomass significant to the distribution of the yield and characterization of the liquid oil but also the operating conditions were crucial. Gozan et al. [17] studied the effect of high temperatures of 500 to 700˚C and found that the optimum yield of liquid oil was obtained at 600˚C. Cardoso et al. [18] investigated the effect of temperatures (400–700˚C) and additives ($ZnCl_2$ and $MgCl_2$) on the distribution of chemical compounds in the liquid oil from the pyrolytic reaction. They found that the liquid oil could be used to produce a fuel with good ignition.

A one-dimensional gas chromatography–mass spectrometry (GC-MS) is usually employed to analyze chemical compounds in the liquid oil from pyrolytic processes and the final product. Further insight into chemical compounds could be achieved by adapting the py-GC-MS technique [19], TG-FTIR, and TG-MS [20]. Yan et al. [21] studied the chemical compounds extracted from tobacco leaves and stems. They reported that most aromatic components were recovered at temperatures below 350˚C and fragmented into small molecules with increasing temperature due to secondary decomposition. Other specific chemical compounds such as nitrogen-containing compounds [22], nicotine [23,24], and phenols [25,26] have been studied. One dimensional GC has been used for many years; however, there remain minor shortcomings in obtaining the best compound separation using this technique. Increasing the resolution and detectability of liquid oil could be obtained using higher-dimensional GC techniques [27], allowing for the detection of a higher number of compounds. A 2-dimensional, GCxGC/TOF MS analysis is believed to be more efficient in identifying more organic compounds in the organic fraction than traditional GC-MS. Even though there have been several studies on tobacco bio-oils, 2-dimensional GC analysis of bio-oils is still scarce.

In this work, the yield and characterization of bio-oil, including aqueous and organic phases, was studied. The effects of temperature, blade rotation speed, and particle size through the ablative reactor were investigated. The elemental and chemical compositions of the organic phase of the liquid oils were analyzed by GCxGC/TOF MS and nuclear magnetic resonance (NMR).

## 2. Experimental

### 2.1 Sample materials

Residues from tobacco harvested and processed at the Tobacco Authority of Thailand 's Denchai redrying plant were used as the raw material in this work. They contained 8.2–15.7% celluloses, 7.5–7.6% hemicelluloses, and 7.1–9.1% lignin [2]. The samples were cut in lengths

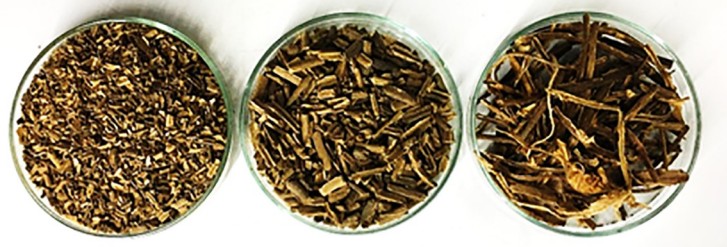

**Fig 1. Tobacco residues with particle sizes 0.5 cm, 1 cm, and 3–5 cm.**

of 0.5 cm, 1 cm, and 3–5 cm, as shown Fig 1, dried at 60°C for 8 h, and stored in Ziploc bags. This feedstock was used to generate the oil with pyrolysis.

## 2.2 Experimental procedure

About 50 g of dried feedstock was introduced into a rotating blade ablative reactor under vacuum at approximately −10.7 kPa (gauge) for each experiment. The pyrolysis setup was adapted from that used in our previous work [2]. In the pyrolytic chamber, the feedstock was heated during effective ablation with four asymmetric blades, which were rotated and pressed against a hot plate. The pyrolytic temperatures varied: 450, 500, 550, and 600°C, while the residence time was fixed at 10 min. During the biomass ablation, volatiles released from the biomass decomposition could be extracted from the reactor; therefore, secondary reactions of the volatiles were prevented. The speed of the blades was varied at 1, 8, and 10 rpm. The pyrolysis liquid products consisting of the organic and aqueous phases were obtained using a dry ice condenser. The yield measurements were replicated at least three times for each condition in order to guarantee reproducibility, and the results were averaged. The collected liquid was weighed after removing the dry ice. The aqueous phase was separated from the organic phase, while the organic phase that adhered to the condenser was dissolved with acetone. Then, both fractions of the oil were stored in enclosed glass bottles in the dark at 0–2°C to avoid any change in the physical properties or chemical composition. The organic phase was further analyzed. The percentage of the oil products was calculated from the material balance relative to the mass of the sample feed.

$$Y_L = \frac{M_L}{M_F} \times 100 \tag{1}$$

where $M_L$ and $M_F$ are the masses of the liquid bio-oil product and the biomass feed, respectively, and $Y_L$ is the yield of the liquid bio-oil.

## 2.3 Feedstock and product analysis

The samples and products were analyzed for proximate and ultimate composition as well as calorific value. In the proximate analysis, the moisture was determined according to ASTM E871. The ash content was inferred following the standard ASTM D1105. The volatile matter was determined using an oven at 980 ± 10 °C, and approximately 1 g of the sample was placed inside of the preheated oven for 10 min. Finally, the sample was removed from the oven, cooled in a desiccator, and the residual masses were determined. Proximate analyses were performed in triplicate for each sample. The ultimate analysis for carbon, hydrogen, nitrogen, and oxygen was carried out using the ASTM D5373 standard, and the sulfur content was determined using ASTM D4239.

[1]H and [13]C NMR were used to analyze the functional groups of the bio-oil samples in deuterated acetone with a Bruker 400 UltraShield NMR that scanned the bio-oil samples at each operating temperature. The [1]H NMR was operated at 400 MHz and 21˚C. The result was represented with a spectral width of 8278 Hz and a resolution of 0.126 Hz. The [13]C NMR was operated at 100 MHz and 22˚C, and also used the [1]H-BB probe of 5 mm BBI with a spectral width of 23980 Hz and a resolution of 0.366 Hz. The NMR spectra were evaluated in Mestre-Nova and were accurately standardized to the acetone reference peak. The chemical shifts scanned during NMR were 0–16 ppm for [1]H and 0–220 ppm for [13]C. The peak results were integrated and then normalized to the total area for representation.

The GCxGC/TOF-MS instrument used consists of a liquid nitrogen quad-jet modulator and a CTC Combipal autosampler, operated in electron ionization mode with an energy of 70 eV, mass acquisition in the range of 50–550 m/z at 100Hz, and a detector voltage of 1706 V. The transfer line and injector were kept at 553.15 K, while the ion source was kept at 523.15 K. Two conventional columns were used: a DB-5 (5% phenyl and 95% dimethylpolysiloxane) with a length of 60 m, an internal diameter of 0.25 mm, and film thickness of 0.1 μm; and DB-17MS (50% phenyl and 50% dimethylpolysiloxane) with a length of 2.15, an internal diameter of 0.18 mm, and film thickness of 0.18 μm. The DB-5 column started at 40˚C for 4 min and heated at a rate of 4˚C to 280˚C, remaining at this value for 15 min. The second column was maintained at 10˚C above the temperature of the first column. The modulation period was 7 s, and the hot pulse was 40% of the modulation period.

The relative area percent for each chromatographic peak was employed as a semi-quantitative approach to evaluating the contribution of each compound area to the total area and for comparison between the different pyrolytic temperatures of the bio-oil samples. The sum of all peak areas was considered to be 100% of the sample. The peak areas related to solvent and column bleed were ignored.

## 3. Results and discussion

### 3.1 Product distribution and yields

During pyrolysis, many factors contribute to the yield and character of the products. From the configuration of the reactor, three parameters associated with the thermal degradation of raw biomass are the temperature, particle size, and speed of the rotating blade. Fig 2 shows the

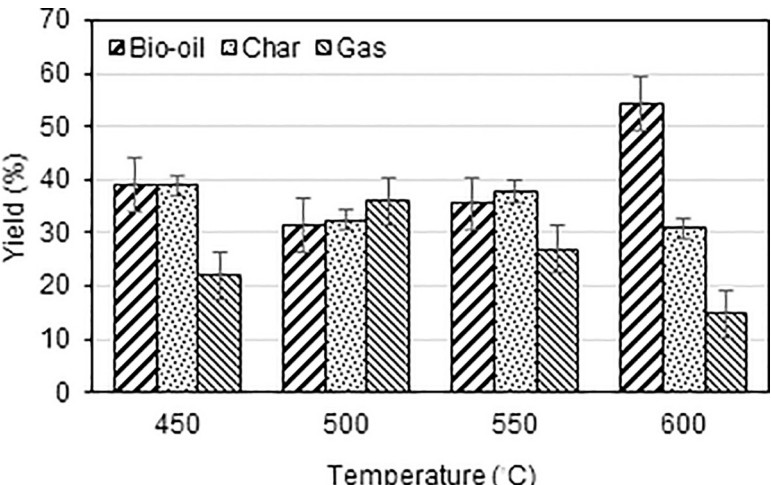

**Fig 2. Product distribution at a particle size of 1 cm and blade speed of 10 rpm.**

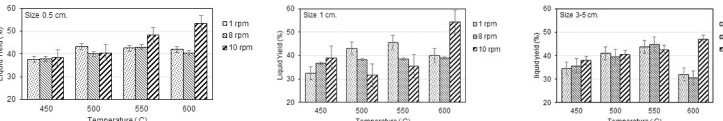

**Fig 3. The effect of particle sizes of (a) 0.5 cm, (b) 1 cm, and (c) 3–5 cm on the liquid yields at varying temperatures and rotation speeds.**

distribution and yield of the pyrolysis products. The highest bio-oil yield of 54.5% w/w was obtained at 600˚C, in similar range to that reported in previous research [28], and the highest char yield was received at 450˚C. As anticipated, lowering the temperature tended to result in higher char yield [29,30]. Nonetheless, the char yields at 500 and 550˚C might have been affected by other factors, such as particle size and blade speed, that directly relate to heat transfer.

As far as the particle size was concerned, the results from the smallest particle size (0.5 cm) indicate that the oil yield was affected by varying temperature and rotating speed, as shown in Fig 3(A). At low and moderate pyrolytic temperatures, the temperature change did not affect the oil yield. At a relatively high temperature of 550˚C, higher speeds result in higher yields, from 42% w/w at 1 rpm to about 48% w/w at 10 rpm. A similar trend was observed at 600˚C when 40% oil yield was improved to 55% w/w when the speed was increased from 1 to 10 rpm.

From Fig 3(B), with the moderate particle size of 1 cm, at 450˚C, the oil yield increases with increasing rotating speed from approximately 32% to 37% and 39% w/w with rotating speeds of 1 to 8 and 10 rpm, respectively. The oil yields at moderate temperatures, 500 and 550˚C, have opposite trends, such that the oil yield decreases with increasing speed. At the highest temperature (600˚C), the oil yields were improved at all speeds, reaching a maximum of 54.4% w/w yield at the highest speed.

For the largest particle size considered, Fig 3(C) shows a slight increase in the oil yields with increasing speed at 450–550˚C. The yields were similar to those obtained from other particle sizes at similar temperatures. At 600˚C, the oil yields were lower at rotation speeds of 1 and 8 rpm, but reached a maximum at the highest speed considered.

## 3.2 Proximate and ultimate analysis of the raw biomass and main products

Proximate results, including fixed carbon and elemental components, as well as the estimate of the heating value of the raw biomass, are shown in Table 1. Ultimate analysis results of the raw biomass and the main pyrolysis products are also included. The raw biomass appears to have high volatile matter (59%) and ash (21%) contents. This volatile matter content was slightly lower, and the ash content was much higher than those reported in other works [21,29,31]. For

**Table 1. Proximate* and ultimate** analysis results of raw tobacco residues and the main pyrolysis products.**

| | Moisture (%) | VM (%) | FC (%) | Ash (%) | C (%) | H (%) | O (%) | N (%) | S (%) | HHV (MJ/kg) | Ref |
|---|---|---|---|---|---|---|---|---|---|---|---|
| Raw | 13.01 | 59.45 | 6.82 | 20.72 | 41.74 | 5.00 | 48.05 | 5.12 | 0.09 | 14.53 | This work |
| Raw | 12.30 | 72.75 | 1.69 | 13.26 | 42.39 | 6.49 | 48.48 | 2.23 | 0.41 | 14.98 | [31] |
| Raw | 10.80 | 68.54 | 11.78 | 8.88 | 46.40 | 5.96 | 45.56 | 2.08 | n/a | n/a | [21] |
| Char | n/a | n/a | n/a | n/a | 59.92 | 3.23 | 33.60 | 2.07 | 1.18 | 13.25 | This work |
| Bio-oil | n/a | n/a | n/a | n/a | 75.07 | 9.15 | 11.84 | 3.80 | 0.14 | 30.05 | This work |

* as-received basis

** dry ash-free basis.

elemental analysis, the C and H concentrations of the bio-oil were 75.1% and 9.2%, respectively, whereas the char values were 59.9% and 3.2%, respectively. Both products showed a higher carbon content than the raw tobacco residues. Moreover, the oxygen content in the oil (11.8%) was lower than in the original biomass feedstock (48.1%) and the char (33.6%). This implied that the bio-oil was likely to be a good liquid fuel. However, the amounts of nitrogen and sulfur present in the oil were high, which poses a challenge for the upgrading process to obtain high-quality biofuels or valuable chemicals.

### 3.3 Chemical composition of the bio-oil

**3.3.1 NMR analysis.** NMR analysis was conducted on the oil samples obtained from pyrolysis temperatures of 450, 500, 550, and 600˚C. The chemical shift region was identified using by [1]H protons, and carbon assignments were based on previous reports in the literature [32,33]. The advantage of NMR analysis is that the whole bio-oil sample can be dissolved in a suitable solvent, and a quantitative assessment of the chemical functional groups can be determined by integrating the defined regions of spectra [34]. The [1]H NMR results are shown in Fig 4 and Table 2. Overall, 1.5–3.0 ppm dominated in oils from all temperatures. Especially in the sample generated at 450˚C, the highest area percentage was evident, relating to the aliphatic OH and ketone groups. Increasing temperatures led to lower percentages in the 1.5–3.0 ppm region. At 500˚C, alkane and aromatic groups were found to be a major part of the oil (31.15% and 13.09%) in the regions of 0.5–1.5 and 6.0–8.5 ppm, respectively. For 550 and 600˚C, the alkanes and aliphatic O-H groups, as well as ketone groups, were found to decrease with increasing temperature, whereas alcohol and methylene groups were found to behave in opposite trend.

An overview of the [13]C NMR results is shown in Fig 5, and the percentage within a given chemical shift range is summarized in Table 3. Information on the type of chemical functional groups in the oils is provided. The region of 1–55 ppm corresponds to alkyl hydrocarbons considered for the provision of energy. Alkyl hydrocarbons are of prime interest when the bio-oil

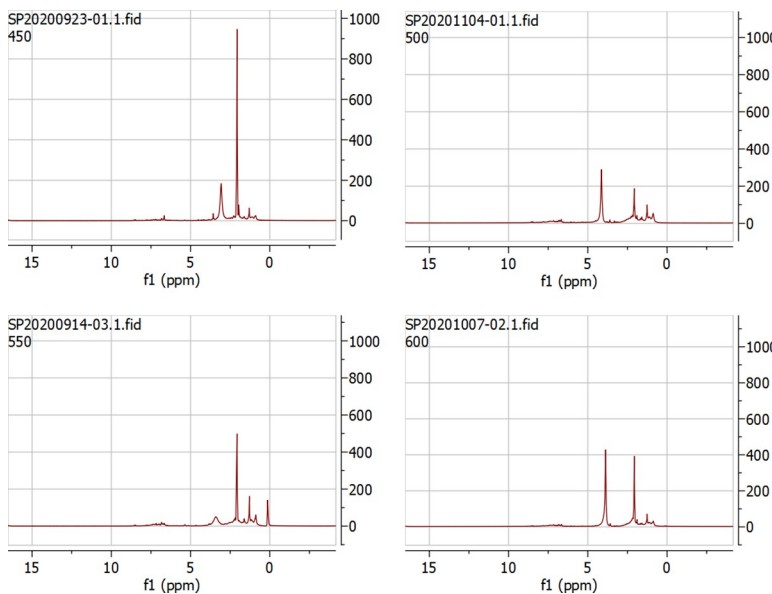

**Fig 4.** [1]H NMR spectra of the organic oils obtained in different temperature (a) 450˚C, (b) 500˚C, (c) 550˚C and (d) 600˚C.

**Table 2. ¹H NMR results for oil generate at different pyrolytic temperatures (in area %).**

| Region ppm | 450˚C | 500˚C | 550˚C | 600˚C | Type of proton |
|---|---|---|---|---|---|
| 0.5–1.5 | 11.64 | 31.15 | 21.45 | 13.50 | alkane |
| 1.5–3.0 | 72.24 | 54.42 | 44.84 | 32.44 | aliphatic OH, ketone |
| 3.0–4.4 | 8.23 | 1.34 | 19.48 | 37.68 | alcohol, methylene |
| 4.4–6.0 | 1.58 | - | 3.18 | 5.02 | methoxy, carbohydrate |
| 6.0–8.5 | 6.29 | 13.09 | 10.80 | 11.35 | (hetero-)aromatic |
| 9.510.1 | - | - | 0.24 | - | aldehyde |

is to be used as fuel [35]. From Table 3, it can be seen that there were two significant groups which include aliphatic and aromatic compounds. The 1–28 ppm region representing short aliphatic chains was most abundant at 600˚C. Higher heat was likely needed to cleave a long chain of polymer from the lignocellulosic biomass into short chain products. It is generally accepted that the short chain aliphatics, such as the alkanes ($C_8$ –$C_{13}$), are important for producing jet and diesel fuels [36]. However, high molecular weight aliphatics in the region of 28–55 ppm were obtained at 500 and 450˚C, with the area percentages of about 68% and 53%, respectively. The region between 95 and 165 ppm represents aromatics, including heteroaromatics, for example, furans and alkenes in the bio-oil. The aromatic content is important for synthetic modification [35]. Carbons corresponding to the aromatic region were abundant at 550˚C, accounting for about 26% of the peak area.

**3.3.2 GCxGC/TOF-MS analysis.** Table 4 shows the results of GCxGC/TOF-MS analysis of oils generated at different pyrolytic temperatures in terms of various organic compounds. The yield of components in the liquid product is expected to vary with temperature because the heat fragments chemical linkages in the biomass to release compounds in the liquid product. By comparing the peak area and the relative abundance in terms of area percentage, the identified bio-oil components were classified into the following main chemical groups: alkanes, alkenes, acids, furans, alcohols, phenolic compounds, nitrogen-containing compounds, benzene derivatives, ketones, and PAHs.

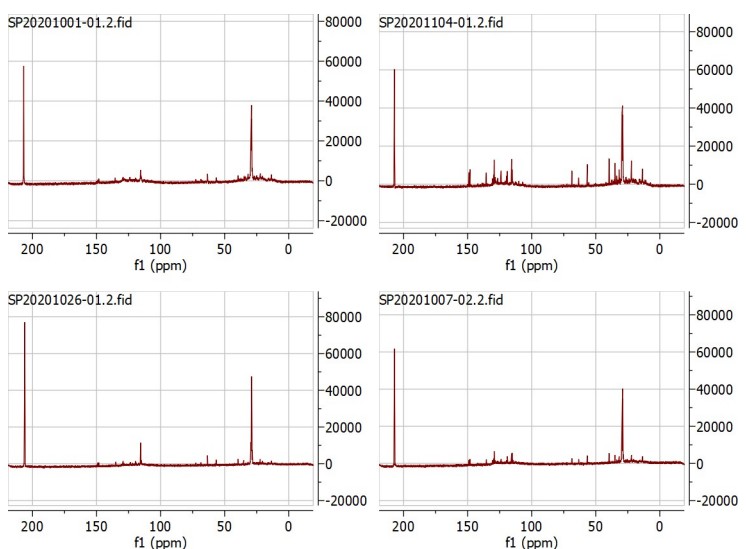

**Fig 5.** ¹³C NMR spectra of the organic oils obtained in different temperature (a) 450˚C, (b) 500˚C, (c) 550˚C and (d) 600˚C.

**Table 3.** $^{13}$C NMR results for oil generated at different pyrolytic temperatures (in area %).

| Region ppm | 450˚C | 500˚C | 550˚C | 600˚C | Type of proton |
|---|---|---|---|---|---|
| 0–28 | 33.67 | 11.99 | 24.59 | 43.01 | short aliphatic |
| 28–55 | 53.19 | 68.22 | 41.54 | 47.65 | long and branched aliphatic |
| 55–95 | - | - | 8.28 | - | alcohol, ether, phenolic-methoxy, carbohydrates |
| 95–165 | 13.14 | 14.87 | 25.58 | 9.33 | aromatic, olefin |
| 165–180 | - | - | - | - | ester, carboxylic acid |
| 180–215 | - | - | - | - | ketone, aldehyde |

The alkanes and the benzene derivatives were the majority of the compounds identified for all temperatures. These alkanes were not found via GC-MS in our previous work [37]. At 600˚C, the alkanes had the highest percentage of peak area, while at 500˚C, the benzene derivatives were the highest. Within the alkanes, decane, tetradecane, hexadecane, and dodecane were among the most abundant components obtained at 450, 500, and 550˚C, whereas toluene was predominant among the benzene derivatives, especially at 500˚C. Compared to those reported in other works, this organic liquid had a high heating value [38,39]. This was because it consisted predominantly of long chain aliphatic and aromatic groups. In addition, it contained small amounts of oxygenated compounds such as ketones, phenols, acids, and furans, as well as alcohols that reduced the heating value.

The phenolic compounds in the pyrolytic oils are a typical product from lignocellulosic biomass, mainly produced from the decomposition of lignin [40,41]. They have been utilized in various household, medical, and solvent products [42]. In this work, higher temperatures tended to produce more phenolic compounds. For example, from 450 to 600˚C, there was a significant increase in the percentage area of phenolics from 3.8% to 10.6%. Ketones are carbonyl compounds used mainly as solvents and intermediates in the chemical industry [43]. The results of the GCxGC/TOF-MS analysis show that a high concentration of the ketone group was found in pyrolysis products at 450 and 550˚C. In particular, methyl Isobutyl ketone was the main component at 450˚C, reaching 5.7%. The furan structure contains a heterocycle of four carbon atoms and one oxygen atom; it primarily results from hemicellulose decomposition and is probably produced from hexoses [44,45]. The furan yield, including furfural decreased with increasing temperature similarly [46]. This trend was opposite to the alkane yield, which suggests a hypothesis to support the possible pathway of the formation of alkanes.

Pyrrole and pyridine,3-(1-methyl-2-pyrrolidinyl), also known as nicotine, were the main nitrogen-bearing compounds of the bio-oils. The group of nitrogen-bearing compounds increased significantly from 10.13% to 15.81% at 450 to 500˚C. Increasing the temperature further resulted in a gradual decline in the concentration to 13.56% at 550˚C, and approximately 9% at 600˚C.

## 3.3 The possible reaction pathway of alkanes

From analysis with 2-dimensional GC, alkanes were found to be rather high, based on peak area percentage. The existence of alkanes in noticeable amount found here was surprising since it was not stressed in previous published reports. It is of interest to propose a possible pathway of alkanes. Alkanes were likely formed from pyrolysis of hemicelluloses since reaction temperatures were not so high in our work. Polysaccharide chain of the hemicelluloses started with depolymerization into oligosaccharides, following the cleavage of monomeric unit including xylan and O-acetyl xylan units. The cleavage and rearrangement of both units were further decomposed to form small molecular compounds, such as furfural, CO, $H_2$, $CH_4$ and short alkanes as well as intermediates [46,47], as shown in Fig 6. Long chain of alkanes may

**Table 4. Distribution of chemical compounds in pyrolytic bio-oil by GCxGC/TOF-MS area percentage.**

| Compound | 450˚C | 500˚C | 550˚C | 600˚C |
|---|---|---|---|---|
| **Alkanes** | **20.529** | **11.504** | **17.215** | **26.928** |
| Butane | | | 0.465 | |
| Octane | 0.499 | | 0.379 | 0.802 |
| Heptane, 2, 5-dimethyl | 3.266 | 1.723 | 2.35 | 4.224 |
| Decane | 3.995 | 2.422 | 3.195 | 6.854 |
| Dodecane | 5.609 | 3.518 | 4.704 | |
| Tetradecane | 2.956 | 1.191 | 2.654 | 5.914 |
| Hexadecane | | 1.405 | | 4.612 |
| Heptadecane | 2.284 | | 2.014 | 0.212 |
| Heneicosane | 1.634 | 1.031 | 1.074 | 3.66 |
| Heptacosane | 0.286 | 0.214 | 0.38 | 0.65 |
| **Alkenes** | **0.123** | **2.29** | **0.197** | |
| Limonene | 0.123 | 0.66 | 0.197 | |
| 1,4-cyclohexadiene,1-methyl- | | 1.63 | | |
| **Acids** | **5.547** | **4.996** | **5.858** | **2.861** |
| Propanoic acid | 2.997 | 0.863 | 1.455 | 2.861 |
| Phosphonic acid, (p-hydroxyphenyl)- | 2.55 | 4.133 | 4.403 | |
| **Furans** | **2.391** | **1.719** | **1.585** | **0.359** |
| Furfural | 0.678 | 0.766 | 1.044 | |
| Furan,2,5-dimethyl- | 0.87 | 0.953 | 0.424 | 0.183 |
| Furan,2-ethyl-5-methyl- | 0.228 | | 0.117 | |
| Furanmethanol | 0.307 | | | |
| Tetrahydrofuran,2,2-dimethyl- | 0.308 | | | 0.176 |
| **Alcohols** | **8.98** | **1.311** | | **0.954** |
| Butanol | 1.311 | 0.382 | | 0.954 |
| 2-pentanone, 4-hydroxy-4-methyl- | 7.669 | | | |
| **Phenols** | **3.843** | **7.782** | **7.769** | **10.624** |
| Phenol | | | | 3.581 |
| Phenol,2-methyl- | 0.525 | 1.007 | 1.091 | 1.061 |
| p-Cresol | 0.693 | 1.762 | 1.893 | 1.476 |
| Phenol,2-methoxy | 0.329 | 0.783 | 0.407 | 0.38 |
| Phenol,2,3-dimethyl | 0.409 | 0.701 | 0.422 | 0.725 |
| Phenol,4-ethyl | 0.919 | 1.502 | 1.874 | 1.77 |
| Phenol,2-ethyl-6-methyl- | 0.191 | 0.788 | 0.586 | 0.597 |
| Phenol,3,4-dimethyl- | 0.062 | 0.165 | 0.182 | 0.123 |
| o-Creosol | 0.028 | | | |
| Phenol,4-ethyl-3-methyl- | 0.164 | | 0.367 | 0.377 |
| Hydroquinone | 0.523 | 0.856 | 0.947 | 0.534 |
| Catechol | | 0.218 | | |
| **N-containing** | **10.133** | **15.811** | **13.558** | **8.972** |
| Butanenitrile | 0.704 | 0.867 | 1.303 | |
| Pyrrole | 3.566 | 4.215 | 3.91 | 2.623 |
| 1H-pyrrole, 3-methyl | 1.154 | 1.963 | 0.806 | 1.741 |
| 1H-pyrrole, 1-methyl | 0.789 | 0.849 | | |
| Indole | 0.398 | 1.05 | 0.702 | 0.759 |
| Pyridine,3-(1-methyl-2-pyrrolidinyl)'(s) | 3.522 | 5.37 | 4.663 | 3.849 |
| 2H-Imidazole,2,24,5-tetramethyl- | | 1.497 | | |

(*Continued*)

**Table 4.** (Continued)

| Compound | 450˚C | 500˚C | 550˚C | 600˚C |
|---|---|---|---|---|
| Pyridine | | | 1.022 | |
| 2-propanamine,N-(1-methylethylidene) | | | 1.152 | |
| **Benzene derivatives** | **15.95** | **24.965** | **18.416** | **21.069** |
| Toluene | 9.017 | 14.327 | 11.052 | 10.441 |
| Ethylbenzene | 2.693 | 3.501 | 2.478 | 3.262 |
| O-xylene | 1.739 | 0.988 | 0.923 | 1.059 |
| P-xylene | 0.509 | 2.779 | | 3.155 |
| Styrene | 1.35 | 1.947 | | 1.164 |
| Benzene, propyl | 0.298 | 0.539 | 0.368 | 0.367 |
| Benzene,1-ethyl-4-methyl- | 0.344 | 0.884 | 0.694 | 0.81 |
| Benzene,1,3-dimethyl | | | 2.504 | |
| Mesitylene | | | 0.442 | |
| Benzene,1,2,3-trimethyl | | | | 0.811 |
| **Ketones** | **10.482** | **7.939** | **11.159** | **6.31** |
| 3-Pentanone | 0.964 | 1.337 | 1.629 | 0.645 |
| Methyl Isobutyl ketone | 5.672 | | 0.71 | 1.563 |
| Cyclopentanone | 1.308 | 0.713 | 0.967 | 1.44 |
| 2-Cyclopentane-1-one | 0.834 | 1.42 | 2.741 | |
| 2-Cyclopentane-1-one, 2-methyl | 0.858 | 1.428 | 1.64 | 0.897 |
| 2-Cyclopentane-1-one, 3-methyl | | 0.736 | 1.004 | 0.482 |
| 2-Cyclopentane-1-one, 2,3-dimethyl | 0.846 | 0.741 | 0.949 | 0.775 |
| 2-Cyclopentane-1-one, 3,4-dimethyl | | 0.57 | 1.222 | 0.508 |
| 1,2-ethanediol,diacetate | | 0.994 | 0.297 | |

have likely been formed by polymerization and recondensation of short alkanes and intermediates during low temperature quenching in a condenser.

## 4. Conclusions

In this work, vacuum ablative pyrolysis of tobacco residues was performed to generate bio-oils. The effects of reaction temperature, particle size, and rotation speed on the pyrolytic

**Fig 6. A possible pathway for conversion of hemicelluloses into alkanes, adapted from [46].**

products were considered. The maximum bio-oil yield of 55% w/w was achieved at 600˚C, particle size of 1 cm, and a blade speed of 10 rpm. The bio-oil obtained was of good quality with high carbon and hydrogen contents and high calorific value. The GCxGC/TOF-MS analysis indicated that the oil was comprised mainly of alkanes and benzene derivatives, which are useful components for upgrading to diesel fuel. The organic bio-oil also contained nitrogen-containing compounds such as nicotine. The $^{13}$C NMR analysis results confirmed the GCxGC/TOF-MS results of long aliphatic chains, the alkanes related to high carbon content. A reaction pathway for these alkanes was proposed: they were possibly converted from decomposition of hemicelluloses.

## Supporting information

**S1 Fig. Dispersion graphics of organic oil in different temperature (a) 450˚C, (b) 500˚C, (c) 550˚C and (d) 600˚C.**
(DOCX)

**S2 Fig. Three-dimensional diagrams of organic oil in different temperature (a) 450˚C, (b) 500˚C, (c) 550˚C and (d) 600˚C.**
(DOCX)

## Author Contributions

**Conceptualization:** Nakorn Tippayawong.

**Formal analysis:** Nattawut Khuenkaeo, Sanphawat Phromphithak, Thossaporn Onsree.

**Funding acquisition:** Nakorn Tippayawong.

**Investigation:** Nattawut Khuenkaeo, Sanphawat Phromphithak, Thossaporn Onsree.

**Methodology:** Nattawut Khuenkaeo, Sanphawat Phromphithak, Thossaporn Onsree.

**Supervision:** Nakorn Tippayawong.

**Writing – original draft:** Nattawut Khuenkaeo, Thossaporn Onsree.

**Writing – review & editing:** Salman Raza Naqvi, Nakorn Tippayawong.

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
