## [Decision Letter · Decision Letter 0]

20 Apr 2021

PONE-D-21-08473

Yields and characterization of bio-oils from fast pyrolysis of tobacco processing wastes in an ablative reactor under vacuum

PLOS ONE

Dear Dr. Tippayawong,

Thank you for submitting your manuscript to PLOS ONE. After careful consideration, we feel that it has merit but does not fully meet PLOS ONE’s publication criteria as it currently stands. Therefore, we invite you to submit a revised version of the manuscript that addresses the points raised during the review process.

We look forward to receiving your revised manuscript.

Kind regards,

Abhay K. Pandey

Academic Editor

PLOS ONE

Additional Editor Comments:

Your MS Needs major revisions based on reviewers' comments. In addition, I have also gone through the MS and found that your MS must be read by a native language editor in order to improve the language of the MS. Also please improve the discussion proving recent references. 

Journal Requirements:

2. Please provide further details on the source of the tobacco residues, such that similar samples could be obtained by another researcher seeking to reproduce your work.

Reviewers' comments:

Reviewer's Responses to Questions

**Comments to the Author**

1. Is the manuscript technically sound, and do the data support the conclusions?

Reviewer #1: Partly

Reviewer #2: Yes

2. Has the statistical analysis been performed appropriately and rigorously? 

Reviewer #1: Yes

Reviewer #2: Yes

3. Have the authors made all data underlying the findings in their manuscript fully available?

Reviewer #1: Yes

Reviewer #2: Yes

4. Is the manuscript presented in an intelligible fashion and written in standard English?

Reviewer #1: Yes

Reviewer #2: Yes

5. Review Comments to the Author

Reviewer #1: Manuscript review

Manuscript number: PONE-D-21-08473

Manuscript title: Yields and characterization of bio-oils from fast pyrolysis of tobacco processing wastes in an ablative reactor under vacuum

Summary:

Vacuum pyrolysis of tobacco residues was performed at various temperatures (400-600 degC) and ablative blade rotation speeds. Product analysis is reported with surprising quantities of alkanes.

General comments:

The paper has many standard experimental analyses seen in pyrolysis literature and there are some surprising results showing alkanes- though without a mass balance it is not clear how much. The biggest weaknesses of the paper are the following:

1) There is no clear hypothesis.

2) Relevant literature is not cited – notably, there are many papers out there regarding vacuum pyrolysis. Further, have other researchers found alkanes to be present in tobacco pyrolysis oils? The manuscript does not discuss if this is a new finding or not.

Location Comment

Introduction The premise of the paper is vacuum pyrolysis, which is a very specialized technology. However you did not cite a vast body of research which has explored how vacuum pyrolysis changes the product spectrum. Please add a paragraph explaining this choice.

Introduction What is the scientific question and hypothesis? Stating a list of experiments is not a very good story, so to speak.

Experimental procedure Please provide a figure illustrating your unique reactor configuration.

Experimental procedure It is so good that you ran the experiments 3x. Nice job.

Results Please compare your results to at least some of the many, many vacuum pyrolysis papers available in the literature. E.g. Garcia-Perez et al. 2002.

Table 1 Please report the ash content and proximate analysis of the char produced from this work. I am very curious what char looks like from this feedstock under vacuum pyrolysis.

3.3.1 NMR analysis Where are your spectra? Tables 2 and 3 should be connected to a figure with overlayed data.

Table 4 Area percentage is a good start. Do you have an idea of mass balance from this reactor? GCxGC only captures monomer and dimer compounds, but it is well known that vacuum pyrolysis produces a large amount of lignin and sugar derived oligomers, as seen in the literature.

Fig. 4 This reaction scheme does not seem likely. Hydrogenation needs either a metal-acid catalyst or very high temperatures. All of the hydrocarbons come from xylose?

General What is the maximum temperature of the hot points in the reactor?

Reviewer #2: Authors have performed vacuum ablative pyrolysis of tobacco residues to generate bio-oils and characterized it through three-dimensional gas chromatography/time-of-flight mass spectrometric analysis and NMR. The idea and experimental plan seem interesting, but manuscript required some corrections as given below pointwise:

In title, please change “yields” as “production”.

In abstract section, please include 1-2 line of background and 1-2 line of conclusion.

At page no. 1, kindly provide reference for the statement (In the north……….via pyrolysis) given in line 27-30.

At page no. 4, in line no. 108, instrument name “Bruker 400 UltraShield” is incomplete, correct is as “Bruker 400 UltraShield NMR”.

In tables, none of the table shows any statistical analysis. Kindly, include statics.

At page no. 6, in line line no. 153-154, i. e. “At the highest temperature (600 ºC)…..54.4% w/w yield at the highest speed”. These results have not been compared with the existing literature. Kindly, compare and discuss your results.

Kindly, include the desired characteristics of efficient bio-oil somewhere in the manuscript.

The compositional analysis of tobacco residues for cellulose and hemicellulose should be included in the manuscript.

6. PLOS authors have the option to publish the peer review history of their article (what does this mean?). If published, this will include your full peer review and any attached files.

Reviewer #1: No

Reviewer #2: **Yes: **Dr. AJAY KUMAR PANDEY

---

## [Author Response · Author response to Decision Letter 0]

2 Jun 2021

Reviewer # 1

The premise of the paper is vacuum pyrolysis, which is a very specialized technology. However, you did not cite a vast body of research which has explored how vacuum pyrolysis changes the product spectrum. Please add a paragraph explaining this choice. 

ANS: The main concept of this work is “ablative pyrolysis”. We included several reviews (refs 2-8) of the related subjected, large body of references on tobacco pyrolysis was evident and included. Ablative pyrolysis under vacuum also mentioned as suggested, page 2, lines 37-44.

What is the scientific question and hypothesis? Stating a list of experiments is not a very good story, so to speak. 

ANS: Our scientific questions and hypotheses are that: (i) “Does ablative pyrolysis offer benefit in converting tobacco residues into bio-oils ?” and “Can we gain further insights into bio-oil composition with GCxGC ? These were in fact the sequence of our presentation in the introduction, and they were novelty/new contributions offered by this work.

Please provide a figure illustrating your unique reactor configuration. 

ANS: Our reactor setup is adopted from that used in reference [27]. It was a minor modification. We did not repeat it here but we referred to it instead.

Please compare your results to at least some of the many, many vacuum pyrolysis papers available in the literature. E.g. Garcia-Perez et al. 2002. ANS: Comparison made as suggested, page 6, line 153-154.

Please report the ash content and proximate analysis of the char produced from this work. I am very curious what char looks like from this feedstock under vacuum pyrolysis. 

ANS: In this work, we focused mainly on the liquid product of fast pyrolysis – bio-oil. Proximate and ultimate analyses of char were not conducted. A pictures of the char is attached.

Where are your spectra? Tables 2 and 3 should be connected to a figure with overlayed data. 

ANS: NMR spectra added as Figures 4 and 5 next to Tables 2 & 3, as suggested

Area percentage is a good start. Do you have an idea of mass balance from this reactor? GCxGC only captures monomer and dimer compounds, but it is well known that vacuum pyrolysis produces a large amount of lignin and sugar derived oligomers, as seen in the literature. 

ANS: Liquid yields are the range of 33-54% w/w. For each component identified by GC, we only had area percentage data. To work out the amount of each chemical, we would need to compare the area percentage data against those from standard chemicals of known amount. Main monomeric & dimeric compounds in the liquid products were detected by normal GC (as seen in the literature). They were also detected here in this work by GC x GC. But with GC x GC, many more monomer and dimer compounds were detected, compared to normal GC. For the heavy components - lignin and sugar derived oligomers, normal GC and GCxGC are not usually able to detect them. Other techniques (such as Gel Permeation chromatography or Ultra-high-performance liquid chromatography/high-resolution multiple-stage tandem mass spectrometry) may be used (Takada et al, 2004; Prothmann et al., 2018).

Takada et al (2004) Journal of Wood Sciences, 50: 253–259, DOI 10.1007/s10086-003-0562-6.

Prothmann et al. (2018) Analytical and Bioanalytical Chemistry, 410: 7803–7814, DOI 10.1007/s00216-018-1400-4

This reaction scheme does not seem likely. Hydrogenation needs either a metal-acid catalyst or very high temperatures. All of the hydrocarbons come from xylose? 

ANS: We agreed with your suggestion, therefore, the possible pathway was revised and updated, page 12, line 279-288 and Fig. 6.

What is the maximum temperature of the hot points in the reactor? 

ANS: 600 oC

Reviewer # 2

In title, please change “yields” as “production”. 

ANS: Changed as suggested

In abstract section, please include 1-2 line of background and 1-2 line of conclusion. 

ANS: Added as suggested, page 1, Lines 12-13, 21-22.

At page no. 1, kindly provide reference for the statement (In the north……….via pyrolysis) given in line 27-30. 

ANS: Added as suggested, page 2, Line 34.

At page no. 4, in line no. 108, instrument name “Bruker 400 UltraShield” is incomplete, correct is as “Bruker 400 UltraShield NMR”. 

ANS: Changed as suggested, page 5, Line 122.

In tables, none of the table shows any statistical analysis. Kindly, include statics. 

ANS: Those data are from analysis based from research instruments. Their measurements uncertainty depends on sample preparation skills of research scientists and instrument resolution. They were reported to be between 2 to 5% with repeated analyses for Table 1, proximate and ultimate analyses. While those in Tables 2,3 & 4 from GCxGC and NMR depended on measurement resolution.

At page no. 6, in line line no. 153-154, i. e. “At the highest temperature (600 ºC)…..54.4% w/w yield at the highest speed”. These results have not been compared with the existing literature. Kindly, compare and discuss your results. 

ANS: Added as suggested, page 6, Lines 153-154.

Kindly, include the desired characteristics of efficient bio-oil somewhere in the manuscript. 

ANS: Bio-oil is a very complex mixture of light and heavy oxygenated HC compounds. It is characterized by its low heating value, high corrosiveness, high oxygen and water content and high viscosity. Prior to its successful utilization, upgrading may be needed. Nonetheless, in terms of proximate analysis and energy content, values shown in this work (Table 1) could provide useful indicator for bio-oil, but giving “desired” characteristics of “efficient” bio-oil may be fantasizing. 

The compositional analysis of tobacco residues for cellulose and hemicellulose should be included in the manuscript. 

ANS: Included as suggested, page 3, lines 85-86.

Additional Editorial Comments:

1) Please ensure that your manuscript meets PLOS ONE's style requirements, including those for file naming 

ANS: ok

2) Please provide further details on the source of the tobacco residues, such that similar samples could be obtained by another researcher seeking to reproduce your work. 

ANS: Provided as suggested, page 3, lines 84-85.

3) Please include captions for your Supporting Information files at the end of your manuscript, and update any in-text citations to match accordingly. 

ANS: ok

4) Your MS Needs major revisions based on reviewers' comments. In addition, I have also gone through the MS and found that your MS must be read by a native language editor in order to improve the language of the MS. Also please improve the discussion proving recent references. ANS: Our MS was initially checked by a English Editing services, Cambridge Proofreading LLC, prior to submission.

---

## [Decision Letter · Decision Letter 1]

29 Jun 2021

Production and characterization of bio-oils from fast pyrolysis of tobacco processing wastes in an ablative reactor under vacuum

PONE-D-21-08473R1

Dear Dr. Tippayawong,

We’re pleased to inform you that your manuscript has been judged scientifically suitable for publication and will be formally accepted for publication once it meets all outstanding technical requirements.

Kind regards,

Abhay K. Pandey

Academic Editor

PLOS ONE

Additional Editor Comments (optional):

The authors addressed reviewer comments

Reviewers' comments:

Reviewer's Responses to Questions

**Comments to the Author**

1. If the authors have adequately addressed your comments raised in a previous round of review and you feel that this manuscript is now acceptable for publication, you may indicate that here to bypass the “Comments to the Author” section, enter your conflict of interest statement in the “Confidential to Editor” section, and submit your "Accept" recommendation.

Reviewer #2: All comments have been addressed

2. Is the manuscript technically sound, and do the data support the conclusions?

Reviewer #2: Yes

3. Has the statistical analysis been performed appropriately and rigorously? 

Reviewer #2: Yes

4. Have the authors made all data underlying the findings in their manuscript fully available?

Reviewer #2: Yes

5. Is the manuscript presented in an intelligible fashion and written in standard English?

Reviewer #2: Yes

6. Review Comments to the Author

Reviewer #2: Authors have addresses all the queries in impressive way, therefore I recommend to accept this manuscript for publication.

7. PLOS authors have the option to publish the peer review history of their article (what does this mean?). If published, this will include your full peer review and any attached files.

Reviewer #2: No

---

## [Editor Report · Acceptance letter]

5 Jul 2021

PONE-D-21-08473R1 

Production and characterization of bio-oils from fast pyrolysis of tobacco processing wastes in an ablative reactor under vacuum 

Dear Dr. Tippayawong:

I'm pleased to inform you that your manuscript has been deemed suitable for publication in PLOS ONE. Congratulations! Your manuscript is now with our production department. 

Kind regards, 

on behalf of

Dr. Abhay K. Pandey 

Academic Editor

PLOS ONE